# The Influences of Sports Psychological Capital to University Baseball Athletes’ Life Stress and Athlete Burnout

**DOI:** 10.3390/bs13080617

**Published:** 2023-07-25

**Authors:** Meng-Hua Yang, Kai-Feng Hsueh, Chia-Ming Chang, Huey-Hong Hsieh

**Affiliations:** 1Department of Physical Education, Health & Recreation, National Chiayi University, Chiayi 621, Taiwan; avon@mail.ncyu.edu.tw (M.-H.Y.);; 2Department of Leisure and Sport Management, Cheng Shiu University, Kaohsuing 833, Taiwan; marat1014@gmail.com; 3Noah Global Solutions, 610 Lawrence Road, Lawrenceville, NJ 08648, USA

**Keywords:** sports psychological capital, athlete burnout, life stress, university athlete students

## Abstract

Previous studies suggested that athletes’ psychological capital level is related to life stress and burnout. Therefore, the purpose of this study was to explore the influences of university baseball athletes’ psychological capital on their life stress and burnout and provide practical suggestions for athletes and coaches to reduce their life stress and burnout. In this study, we used athletes’ control variables (grade, year of training experience, and training days per week) and psychological capital (self-efficacy, hope, optimism, and resilience) to predict their life stress and burnout. A total of 428 division I baseball athletes from 16 teams of the national college baseball sports league in Taiwan participated in this survey, with a return rate of 89.2%. Partial least squares structural equation modeling was used to test the relationships among the above-mentioned variables. The results showed that the athletes demographics such as grade (β = 0.03, *p* > 0.05) and years of baseball training experience (β = 0.00, *p* > 0.05) had no significant influences on athlete burnout, while the days of baseball training per week (β = 0.32, *p* < 0.05) had a positive influence on athlete burnout. As for psychological capital, self-efficacy (β = −0.09, *p* < 0.05), hope (β = −0.27, *p* < 0.05), and optimism (β = −0.20, *p* < 0.05) had negative influences on life stress, while resilience (β = −0.07, *p* > 0.05) had no significant influences on life stress. Hope (β = −0.20, *p* < 0.05) had negative influences on athlete burnout, while self-efficacy (β = −0.00, *p* > 0.05), optimism (β = −0.06, *p* > 0.05), and resilience (β = −0.01, *p* > 0.05) had no significant influences on athlete burnout. Life stress (β = 0.52, *p* < 0.05) had significant influences on the burnout. Based on our research findings, suggestions were made to reduce the athletes’ life stress and athlete burnout.

## 1. Introduction

Athlete burnout had drawn significant attention in sports psychology. Athlete burnout may increase serious mental burden in athletes and gradually lead to quit. Hence knowing its causes and ways to prevent it may contribute to the athletes’ success and endurance. Raedake [1] defined athlete burnout as a syndrome of physical/emotional exhaustion, sport devaluation, and reduced accomplishment. With the increase of athlete burnout, athletes exhibit anxious, nervous, disturbed, and depressed status, which may affect their performance and may even lead them to withdraw from playing [2,3].

Stress is well recognized as the main cause of athlete burnout [4]. According to Smith [5], athlete burnout develops through a stress-based process influenced by personal and motivational factors. For example, Culter and Dwyer [6] examined 158 US division I collegiate student–athletes’ academic and athletic lives; they found these athletes face multiple stresses from academic and sports performances and worries for the future. Most of them agreed that they were constantly affected by those stresses. Similarly, Holde et al. [7] investigated college student–athletes’ stressors and concluded that college student–athletes’ stressors may come in many forms such as playing time, injuries, discontentment with coaching style, poor academic performance, relationships with teammates, and their win and lose record. Their study also pointed out athletes with different levels or grades all suffered from different levels of stress. Two studies [8,9] focusing on elite student–athletes’ lives had concluded that athletes’ life stress caused burnout. Since baseball is a very popular sport in Taiwan [10], knowing the degree of athletes’ stress affecting their burnout can help athletes and coaches to mitigate athletes’ stress and decrease their burnout. Hence, this study intended to explore the relationships of college student–athletes’ life stress and burnout.

Previous studies showed that sports psychological capital can help athletes coping with stress and lower stress level induced by poor performance [11]. The concept of PsyCap was first developed in the 2000s within the positive psychology movement raised by Luthans, Youssef, and Avolio [12]. They defined PsyCap as “an individual’s positive psychological state of development”. Namely, PsyCap refers to our mental resources and their ability to help us get through tough situations. PsyCap is reported to have benefits for both individuals and organizations. According to Luthans and Larson [13], people with greater PsyCap typically enjoy higher levels of job satisfaction, commitment, and overall wellbeing. In a study of PysCap’s influences on job performances and stress, Abbas and Raja found that people with higher levels of PsyCap can be linked to lower stress and better performances [14]. According to Luthans, Yousself and Avolio [12], PsyCap comprised of four elements: self-efficacy, hope, optimism, and resilience. Hope is defined as one’s willingness to plan for the future and strive toward goals; efficacy is the belief in one’s ability to successfully manage tasks; resilience is one’s ability to “bounce back” following adversity or failure; optimism corresponds to having a positive outlook for the future. With the rise of positive psychology, PsyCap was also applied to the studies of athletes’ mental wellness. Chang and Chi [15] pointed out that the PsyCap theory suggests that athletes can improve their performance and mental wellness through the enhancing of PsyCap. The concept was further elaborated by Chien et al. [11], in which they pointed out that by utilizing personal positive resources, one can be equipped to handle challenges and difficulties. Athletes with high PsyCap can perform steadily even under intense competitive scenarios and can quickly recover from loss and be back to regular training [15]. As Chien et al. [11] pointed out, not only intensive training is important in modern day sports training, but enhancement of athletes’ PsyCap is also important. As sports PsyCap can reduce athlete burnout [16,17,18], it is interesting to explore the PsyCap effects on life stress and athlete burnout. Therefore, the aims of this study were to use sports PsyCap as a predictor to examine its influences on college baseball athletes’ life stress and burnout. Based on the two afore-mentioned purposes, the research framework can be described as follows.

### 1.1. The Relationships of Athletes’ Demographics and Athlete Burnout

Some studies have used demographic variables as control variables to examine the influences of independent variables on predicted variables [18,19]. Reasons for that were to control the variables to better estimate the independent variables. Some studies also indicated that several control variables are related to athlete burnout. For example, different grades’ athletes exhibited different levels of burnout [20,21]. Specifically, Yang et al. [21] reported that in junior high school, lower grade’s athletes had higher athlete burnout than higher grade’s athletes. As for training experience, studies suggested that athletes with different training experience exhibit different levels of burnout [22,23,24,25]. For example, Yang et al. [23] pointed out that female college baseball players with greater training experience had higher burnout. As for training frequency, athletes with different training frequency exhibit different levels of burnout [24,25,26]. For example, Cheng and Yang [23] pointed out that judo players with greater training days per week had higher burnout. Based on above mentioned studies, the following hypotheses are proposed:

**Hypothesis 1-1 (H1-1).** 
*College baseball athletes’ grades have negative influences on athlete burnout.*


**Hypothesis 1-2 (H1-2).** 
*College baseball athletes’ years of training experience have positive influences on athlete burnout.*


**Hypothesis 1-3 (H1-3).** 
*College baseball athletes’ training days per week have positive influences on athlete burnout.*


### 1.2. The Relationships of Athletes’ Sports PsyCap and Life Stress

The sources of stress refer to any situation, environment, or stimulation which may increase one’s feeling of tension. Under intense stress, athletes may exhibit the following syndromes: fatigue, hypertension, headaches, depression, and anxiety [8]. Previous studies showed that stress is common among athletes and has negative effects on athletes’ physical and mental health [7,27,28]. According to Holden et al. [7], for student–athletes, the amalgamation of stressors from their dual roles is often a burden rather than a beneficial experience. When athletes confront various stressors, they may not be able to handle situations well with positive attitudes and effective ways. In such cases, if coaches can help athletes promote their PsyCap, they can be better equipped to deal with difficult situations [29]. Mao [30] pointed out that being optimistic reduces stress and people with high self-efficacy traits can turn stress into an affordable challenge. In other words, PsyCap’s optimism and self-efficacy are negatively related to stress. According to Chen and Chi [31], it was found that among college student–athletes, those with higher hope perception have the better coping strategies. In summary, the four elements of PsyCap (i.e., self-efficacy, hope, optimism, and resilience) have negative influences on stress. Drawn from those findings, our study proposed the following hypotheses:

**Hypothesis 2-1 (H2-1).** 
*Self-efficacy has negative influences on the life stress of college baseball student–athletes.*


**Hypothesis 2-2 (H2-2).** 
*Hope has negative influences on the life stress of college baseball student–athletes.*


**Hypothesis 2-3 (H2-3).** 
*Optimism has negative influences on the life stress of college baseball student–athletes.*


**Hypothesis 2-4 (H2-4).** 
*Resilience has negative influences on the life stress of college baseball student–athletes.*


### 1.3. The Relationship of Athletes’ Sports PsyCap and Athlete Burnout

Hobfoll [32] posited the conservation of resources (COR) theory in 1989. The COR basically defines “things that one values, specifically objects, states, and conditions” as resources. COR claims that the loss of these three types of resources drives individuals into certain levels of stress. In order to avoid the loss of the resources, one can take action to protect the resources from being lost. One way of preventing resources loss is resource investment. People can invest resources to prevent future losses. Sports PsyCap is one type of resource that athletes can invest in to avoid stress or burnout [33]. Many studies concluded that victories and losses of the top list athletes not only depends on their skills, and the crucial factor for victories is their mental toughness [34,35]. For example, athletes with higher sports PsyCap are equipped with strong self-confidence and will look for an optimized path to achieve their goals. When facing challenges and difficulties, they can remain optimistic, and they can also recover to normal states from failures [16,17,18]. Most athletes starts training at an early age. The routine training schedule in daily lives may be boring, and the fatigue and frustration during training may increase the athletes’ burnout level [36]. If student–athletes are equipped with better PsyCap, they can be better off dealing with training induced stress and avoid athlete burnout. Positive psychology nowadays is amply applied in enhancing athletes’ PsyCap to reduce athlete burnout. For example, studies found that junior high school physical class student–athletes with higher sports PsyCap have a lower athlete burnout perception [37,38,39]. Yang et al. [40] investigated 388 junior high school physical education class students’ athlete burnout, finding that students who possess higher self-efficacy and optimism traits have lower athlete burnout. Chuang and Tsai [41] explored 232 badminton athletes in a Taiwanese college campaign. The results indicated optimism was negatively related to athlete burnout. Based on the studies related to sports PsyCap and athlete burnout, the following hypotheses were proposed:

**Hypothesis 3-1 (H3-1).** 
*Self-efficacy has negative influences on athlete burnout of college baseball student–athletes.*


**Hypothesis 3-2 (H3-2).** 
*Hope has negative influences on athlete burnout of college baseball student–athletes.*


**Hypothesis 3-3 (H3-3).** 
*Optimism has negative influences on athlete burnout of college baseball student–athletes.*


**Hypothesis 3-4 (H3-4).** 
*Resilience has negative influences on athlete burnout of college baseball student–athletes.*


### 1.4. The Relationship of Athletes’ Life Stress and Athlete Burnout

Smith [42] pointed out that high levels of athlete stress can have a wide range of negative consequences. For example, they can increase the risk of injuries. Even elite athletes suffered from athlete burnout and eventually dropped out of sports at the peak of their career. In a systematic review and meta-analysis related to athlete stress and burnout [28], the findings indicated that burnout is a negative consequence of stress. The correlation coefficients of athlete stress with overall burnout and the three dimensions of burnout ranged from 0.40–0.53. A study from Hamlin [43] investigated 182 young (18–22) elite student–athletes’ stress levels. The study collected the elite athletes’ mood state, energy level, academic stress, sleep quality/quantity, muscle soreness, training load, injury, and illness over a 4-year period. The findings showed that young elite athletes undertaking full-time university studies alongside their training and competition loads were vulnerable to increased levels of stress at certain periods of the year (pre-season and examination time). Silva’s negative- training stress response model [44] posited that during training, appropriate stress is beneficial for skills improvement. However, excessive stress may induce positive and negative adaptations to training stress. Positive adaptation helps athletes to gradually surpass their present training load and turn to the next level. On the other hand, overly stress and training load may cause negative emotion. If the situations persist, athletes may feel unconfident and become anxious, nervous, ill, uncomfortable, afraid, and resistant upon training [45]. From the above-mentioned studies, we concluded that stress is positively related to athlete burnout. Therefore, the following hypothesis is proposed:

**Hypothesis 4 (H4).** 
*Life stress has positive influences on athlete burnout of college baseball student–athletes.*


## 2. Materials and Methods

### 2.1. Participants

This study recruited 450 division I college male baseball players from 16 teams in Taiwan to participate in the study. Questionnaires were distributed through team coaches. A total of 428 copies were collected with a valid return rate of 95.1%. Table 1 presents the descriptive statistics of the participants. Among them, 158 (36.8%) were freshman, 111 (25.9%) were sophomore, 82 (19.2%) were junior, and 77 (18.0%) were senior. As for training experience, most of them had 6–10 years’ training experience (252/58.9%), the second rank is 11–15 years (155/36.2%). As for training days per week, it varied: 115 (25.9%) had three days training per week, 123 (28.7%) had four days, 108 (25.2%) had five days, and 80 (18.7%) had six days.

### 2.2. Instruments

#### 2.2.1. Sports PsyCap Scale

Sports PsyCap scale was established by adopting the PsyCap scale developed by Yang et al. [21]. The scale was originally used to acquire junior high school student–athletes’ perception of sports PsyCap. The scale comprised of 20 items with four sub-construct, namely, self-efficacy, hope, optimism, and resilience. The questions were rephrased for 18–22 years old student athletes and the scale was rated by a 5-point Likert’s scale ranging from “1 = strongly disagree” to “5 = strongly agree”.

#### 2.2.2. Life Stress Scale

This study adopted the college student–athletes’ life stress scale developed by Lu et al. [46]. The scale consists of two dimensions, namely, general life stress (6 items) and sports life stress (6 items). The scale was rated by a 5-point Likert’s scale ranging from “1 = strongly disagree” to “5 = strongly agree”.

#### 2.2.3. Athlete Burnout Scale

The study adopted the athlete burnout scale developed by Lu, Chen, and Cho [47]. The scale comprised of three dimensions: physical and emotional exhaustion (4 items), low accomplishment (4 items), and devaluation of sports (4 items). The scale was rated by a 5-point Likert’s scale ranging from “1 = strongly disagree” to “5 = strongly agree”.

#### 2.2.4. Demographics

In the study, the demographics included grades, years of training experience, and training days per week.

### 2.3. Data Analysis

The descriptive analyses were performed using SPSS for Windows, and the tests of hypotheses were performed using partial least square (PLS) structural equation modeling by using WarpPLS 8.0 developed by Kock [48]. According to Chin [49], PLS can be used for exploratory and confirmatory analyses. PLS benefits from: (1) being distribution-free, (2) requiring only a small sample size, (3) having the ability to process multiple dependent and independent variables simultaneously, (4) handling collinearity, and (5) processing both formative and reflective indicators. In the study, hypotheses include multiple independent and dependent variables, therefore, PLS was employed for hypothesis testing. A significant level of α = 0.05 was used in the study. 

## 3. Results

### 3.1. Measurement Model

The reliability and validity of the study instrument were tested using WarpPLS 8.0 [48], which, under PLS, provides two measures of item reliability: composite reliability and Cronbach’s α. Convergent validity and discriminant validity were conducted to test validity of the instrument according to Fornell and Larcker [50]. The factor loading of all items from the PLS measurement model was greater than 0.70 in all cases, indicating good indicators. Composite reliability and Cronbach’s α values for all scales exceeded the minimum threshold level of 0.70 [50], indicating the reliability of all scales used in the study (Table 2). As for convergent validity, the square root of average variation extract (AVE) of all values exceeded the minimum threshold level of 0.70 [50], indicating the reliability of all scales used in the study (Table 2). Fornell and Larcker’s test for discriminant validity revealed relatively high variances extracted for each factor compared to the interscale correlations, which was an indicator of the discriminant validity of the four constructs (Table 2).

### 3.2. Hypothesis Test Results

The study used WarpPLS 8.0 [48] to perform hypothesis testing. The results are shown in Figure 1. Following, the descriptions of the test results:

As for the influences of control variables on athlete burnout (H1-1–H1-3), the test results showed that only the days of training per week had significant influences on athlete burnout (path coefficient = 0.32, *p* < 0.05). The more training days, the greater the athlete burnout.

As for the influences of sports PsyCap on life stress (H2-1–H2-4), the results showed self-efficacy, hope, and optimism had significant negative influences on life stress.

As for the influences of sports PsyCap on athlete burnout (H3-1–H3-4), the results showed hope had significant negative influences on athlete burnout.

As for the influences of life stress on athlete burnout (H4), the results showed life stress had significant positive influences on athlete burnout. The path coefficient is greater than 0.50, which indicates high relations between them.

From the results, only hope had significant effects on burnout. Hence life stress mediated the effects on hope to burnout. The mediation effect can be tested using the VAF coefficient suggested by Hair et al. [51] in which VAF = (indirect effect)/(indirect effect + direct effect). The VAF = (−0.27 × 0.52)/(−0.27 × 0.52 + (−0.15)) = 0.48, which indicated a partial mediation [51].

### 3.3. Explanatory Power

Explanatory power, usually denoted as R^2^, indicates the percentage of the change occurring in the dependent variable that is explained by the change in the independent variables. It assesses how well a model explains and predicts future outcomes. Thus, high R^2^ produces precise prediction [49]. As shown in Figure 1, sport PsyCap induced 33% variation on life stress. Control variables (grade, years of training experience, training days per week) and sports PsyCap together with life stress explained 84% variation of athlete burnout, indicating good predicting power.

## 4. Discussion

In this section, we will compare study results with related studies and provide practical suggestions for athletes and coaches.

### 4.1. Demographic Variables and Athlete Burnout

As for demographic variables, our results showed that grades and years of training experience had no influences on athlete burnout, which is not consistent with previous studies [21,23]. As for grades, Yang et al. pointed out that junior high school student–athletes’ grades relate to burnout negatively [21]. As for junior high school athletes, their physical fitness and motor skills vary with grades. Junior high school students undergo physiological growth. As they get older, their physical status grow to mature status. Besides, they get more improvements in motor skills from training. Therefore, younger athletes might have more athlete burnout than older athletes. On the other hand, college athletes’ physical fitness and motor skills are pretty much the same and stable. Therefore, grade may not be a control variable for athlete burnout. As for years of training experience, the study found that college baseball players’ training experience had not influences on athlete burnout. This is different from what reported by Yang et al. [23] in which the study subjects are female players, while our study focuses on male players. Therefore, it can be concluded that male college baseball players’ years of training experience has no influences on athlete burnout. As for the training days per week, our study found that the training days per week had negative influences on athlete burnout. This is consistent with some previous studies [24]. According to descriptive statistics, 43.5% of athletes had 5- or 6-days training per week. Since baseball is a very competitive sport in Taiwan, most baseball student athletes desire to join professional baseball teams [10]. Hence, training is intensive. However, as McGuff and Little [52] pointed out that the best training approach is “high intensity and low frequency”, in that way, athletes get enough rest for recovery. Five- or six-day’s training per week might cause athletes’ physical and mental burden, hence induce stress and fatigue [53]. Therefore, the better approach is to adopt structured training plans, which comprise of high intensity, low frequency training. In that way, athletes are allowed to have adequate resting and recovery periods and athlete burnout can be reduced.

### 4.2. Sports PsyCap and Life Stress

For general people, exercise can release stress, but for professional athletes, exercise may be a source of stress. To maintain a high level of performance, college student–athletes are under high courses of physical and mental stress. This study found that three elements of sports PsyCap (i.e., self-efficacy, hope, and optimism) have negative influences on life stress, which is consistent with some previous findings [20,30,31]. Therefore, athletes can invest in those PsyCap resources to release life stress. Coaches can help athletes for PsyCap improvement. For example, by simulating intense moments during contests, trying to help athletes strengthen their self-confidence in these scenarios, athletes’ self-efficacy can be improved, and athletes can be more confident during contests [54]. Furthermore, if possible, psychological counseling can be embedded in the training schedules to help athletes focus on their performances, despite the loss or win during contests [4].

### 4.3. Sports PsyCap, Life Stress and Athlete Burnout

The study found hope has negative influences on athlete burnout, which is consistent with Gustatasson, Hassmen, and Podlog’s findings [55]. Their findings also indicated that hope has negative influences on athlete burnout. Some studies found that the hope intervention courses for athletes could increase athletes’ hope. For example, Chen and Lee [56] used five-weeks’ hope intervention courses for college tennis players before contests. The study found that after the intervention’s courses, these players had higher hope than the control group. The study also asked whether experimental group players like to have similar hope intervention courses before contests and all of them agreed to have such interventions. Similarly, we suggest coaches offer hope intervention courses before contests. 

This study also found that life stress increases athlete burnout. Wang, Lee, and Lu [57] pointed out that college student–athletes are a unique population that requires special attention, as they might experience stress and develop burnout if they do not have the coping resources to deal with both academic and sport demands. College student–athletes’ life stress comes from general life stress and athlete stress. As for general life stress, Lo, Shiau, and Lin [58] suggested that the college student affairs department offer courses (i.e., life education, interpersonal relationship, self- development, etc.) to help athletes fit in college lives. In addition, in the East and Southeast Asian societies, academic performance is highly valued, therefore student–athletes may focus on training and pay less attention to academic performance. Some athletes may also want to become coaches later on as one of their career options. They need to be certified for coaching. Hence, academic performance is also important for them. Therefore, it is suggested that coaches remind athletes to spend some time studying.

As for athlete stress, Hsu [59] pointed out that athletes’ stress mainly comes from contest pressure. As they get more wins, the contests become more competitive. More intense and frequent training is needed to prepare for wins. This could cause athletes’ injuries and the overtraining may increase their burnout and may lead to quitting. Therefore, coaches need to be aware of this issue. Upon planning training schedules, coaches need to be aware of individual differences. Thorough planning according to athletes’ conditions and adjustments of training intensity and frequency should be promptly implemented. Moreover, if athletes get injured upon trainings or contests, enough time for recovery should be given.

## 5. Conclusions and Suggestions 

### 5.1. Conclusions

The study explored the influences of college baseball players’ PsyCap (self-efficacy, hope, optimism, and resilience) on life stress and athlete burnout using athletes’ grades, years of training experience, and training days per week as control variables. The study found that self-efficacy, hope, and optimism have negative influences on life stress. Hope has negative influences on athlete burnout. Moreover, life stress has a positive influence on athlete burnout. As for demographic variables, the study found athletes’ training days per week have negative influences on athlete burnout. 

### 5.2. Suggestions

Based on our findings and discussion, the study provides the following suggestions for college baseball student–athletes and baseball coaches.

As for student–athletes, this study found that sports PsyCap helps the reduction of life stress and athlete burnout. Among sports PsyCap, self-efficacy is the mental state that one believe he/she is capable of doing his/her job/task. To gain more self-efficacy in sports, mastery is important. Moderate self-challenging can increase mastery so to have more confidence. Besides, different competitive scenarios during trainings should be created. The more the practices, the more comfortable athletes can handle different situations during contests and confidence gains. As for hope, athletes are advised to think of valuable goals and find ways to achieve thier goals. As for optimism, athletes are advised to keep a positive mindset and treat situations calmly. For example, if a defensive error does occur, just focus on how it occurs instead of blaming yourself and pay more attention to defense. In these ways, athletes can be better equipped with sports PsyCap [54].

As for college baseball coaches, this study found that the training days per week had negative influence on students’ life stress and athlete burnout. Nearly half of the participants had 5–6 days’ training per week. Therefore, the study suggests that coaches establish well-planned training schedules according to contest schedules, as well as set training and resting schedules, clearly so that athletes get to rest and study, and athlete burnout is reduced. As for hope, coaches can provide some PsyCap intervention courses before contests [57]. In that way, athletes can benefit from the psychological training and increase their PsyCap.

### 5.3. Implications for Future Research

The study explored the sports PsyCap effects on college baseball athletes’ life stress and burnout. Our finding suggested that athletes with higher sports PsyCap had lower life stress and lower sports burnout. The study can serve as a frontier in creating a link between sport PsyCap and sport burnout for future studies. Based on the results, future research can explore the effects of sport PsyCap intervention courses before contests on athlete burnout. It will also be interesting to explore the relationship between training frequency and intensity and life stress and athlete burnout.

## Figures and Tables

**Figure 1 behavsci-13-00617-f001:**
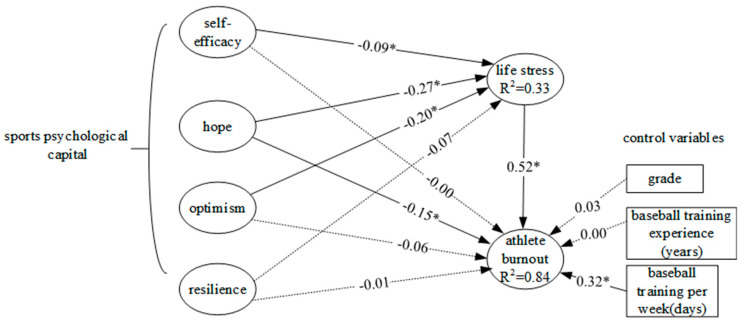
Hypothesis test results of the standardized model parameter estimation. Note: dotted lines represent “path coefficient was not significant”; solid lines represent “path coefficient was significant”. * *p* < 0.05; R^2^: coefficient of determination.

**Table 1 behavsci-13-00617-t001:** Descriptive statistics of participants (*n* = 428).

Variable	Category	*n*	%
Grade	Freshman	158	36.9
	Sophomore	111	25.9
	Junior	82	19.2
	Senior	77	18.0
Training experiences (years)	1–5	16	3.7
	6–10	252	58.9
	11–15	155	36.2
	Above 15	5	1.2
Training days per week	2	2	0.5
	3	115	26.9
	4	123	28.7
	5	108	25.2
	6	80	18.7

**Table 2 behavsci-13-00617-t002:** Reliability, convergent, and discriminant validity of measurement models.

Construct	M	SD	(1)	(2)	(3)	CR ^b^	A ^c^
(1) Sports Psychological Capital	4.05	0.60	0.76 ^a^			0.96	0.96
(2) Life Stress	2.25	0.75	−0.56	0.73 ^a^		0.93	0.92
(3) Athletes Burnout	2.27	0.80	−0.64	0.85	0.78 ^a^	0.95	0.94

Note: M: mean; SD: standard deviation; ^a^ square root of AVE (average variance extracted); ^b^ composite reliability; ^c^ Cronbach’s α.

## Data Availability

Data can be accessed upon request.

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
