# Peer review of "The Influences of Sports Psychological Capital to University Baseball Athletes’ Life Stress and Athlete Burnout"

_behavsci, 2023, doi:10.3390/bs13080617_

Round 1
Reviewer 1 Report
I would like to thank the authors for considering and addressing some of the comments and concerns raised in response to the previous submission. Where changes have been made, the work has been improved. However, it was notable that a substantial number of comments were not addressed in-text, and in some cases no response or rationale was provided in the accompanying document. Specifically, it is my opinion that the following comments were not addressed, either at all or sufficiently well;
1st Review, Abstract: The opening line in the abstract suggests that training demands will be explored as a predictor of stress and burnout, and this is not the case. I would recommend focusing on introducing the concept of stress more explicitly in terms of its relationship to burnout, as you do for PsyCap in the subsequent sentence.
Response 1: Thank you for the comments. We revised the aims of the study as the following: “the purpose of this study was to explore the influences of university baseball athletes’ psychological capital on their life stress and burnout and provide practical suggestions for athletes and coaches to reduce their life stress and burnout.” To make our study goal clearer. Please refer to lines 16- 18.Thank you very much.
2nd review: The concern raised in the original comment has not been addressed, as the opening line (L14 – L15) has not been amended as suggested.
1st Review, Introduction: It would be useful to expand on the description burnout provided to include a definition of each of the three burnout dimensions. In addition, as two different definitions of burnout are provided (i.e. Eades’ and Radeke’s definitions), it should be made clear here which definition will be employed moving forward in the study. To this end, I suggest that Raedeke’s definition is most appropriate, as it is the most commonly employed definition in the athlete burnout literature (e.g. Gustafsson et al., 2017)
Response 3: Thank you for the comments. Ths study mainly focused on athletic burnout and we just focused on the definition of athletic burnout.
2nd review: The concern raised in the original comment has not been sufficiently addressed - Two different definitions of burnout are introduced and the authors need to be clear on which definition they have employed in the study.
1st Review: The introduction to the concept of stress seems to move back and forth between mental health and stress. In my opinion, this paragraph would benefit from a more focused discussion of stress in the context of sport, and its relationship to burnout and psychological capital specifically. Currently, much of this information comes in sections 1.2 and 1.4, but I feel that integrating it into the main body of the introduction would contribute to a more logical flow in this section. As it stands, concepts are being introduced with minimal context, which is confusing for the reader.
2nd review: This point has not been addressed in-text and the author has not provided any direct response.
1st Review: Currently, the main body of the introduction also provides limited insight into our understanding of the relationship between PsyCon and burnout, and I think the following recommendations may help to address this;
1. A definition of the core components of psychological capital would be a useful addition to the introduction. Currently, self-efficacy, hope, optimism and resilience are mentioned but not
defined in this section, with the first definition of self-efficacy is provided in the integrations section. This information would be much more useful earlier in the manuscript.
2nd review: This point has not been addressed in-text and the author has not provided any direct response.
2. In addition, while research exploring the impact of psychological capital on athlete burnout may be relatively limited, it could be useful to refer to existing work exploring this relationship in the job burnout context (e.g. López-Nunez et al., 2020), with a view to providing a stronger rationale for the current study. Similarly, while not explicitly focused on psychological capital, it would be useful to draw on additional existing research exploring the relationship between the individual components of PsyCap and burnout, including work exploring the role of hope (e.g. Gustafsson et al., 2013), resilience (e.g. Vitali et al., 2015) and self-efficacy (e.g. Kocak, 2019) in the experience of athlete burnout.
2nd review: Although the author has not provided any direct response here, Section 1.3 (L159) now focuses on the relationship between PsyCap and burnout.
3) In my opinion, the introduction would be strengthened by incorporating the information from section 1.2 and 1.3 into the main body of the introduction. Doing so would help to address some of 3 the concerns outline above, and would provide a much more logical flow to the introduction, with the full list of hypotheses to follow.
Currently, there appears to be little consideration given to how the three variables of interest might relate to each other when explored in combination. Specifically, the distinct relationships between PsyCap and stress, PsycCap and burnout, and stress and burnout are discussed, but I found it surprising that there was no consideration of potential mediating relationships (e.g. stress as a mediator between PsyCap and burnout). More clearly delineating the link between all three variables would help to provide a stronger rationale for the current study. In addition, it would be useful to be clear as why no mediation analysis was conducted.
Response: thank you for the suggestion. The mediation issues were addressed in the next response. Thank you.
2nd review: The addition of information on mediation in the results section does not address this point as it relates to the introduction.
Methods.
1st Review: In line with the point above, consideration of the existing hypotheses and Figure 1 suggests that would be useful to explore the extent to which stress mediates the relationship between PsyCap and burnout. This information may be readily available from the existing output, or it may be worth considering exploring it further.
Response: Thank you for the comments. From the results, only hope had significant effects on burnout. Hence life stress mediated the effects on hope to burnout. The VAF(Hair, J. F., Hult, G. T. M., Ringle, C. M., & Sarstedt, M. (2014). A Primer on Partial Least Squares Structural Equation Modeling (PLS-SEM), 3rd ed. Thousand Oaks, CA: Sage.)=(-0.27*0.52)/(-0.27*0.52+(-0.15))=0.48 which indicated a partial mediation. We had adressed the mediation in section 3.2. Hypothesis tests. Thank you.
2nd review: This information appears to be included as a foot note in the results section. It would be beneficial to include this in the main body of the text, and to discuss how mediation will be assessed in the methods section. Furthermore, as per the previous comment, exploration of moderating
relationships should also be discussed in the introduction and should be included in the list of hypotheses.
1st Review, Discussion: Broadly speaking, I think the discussion would benefit from providing a broader initial discussion of the key findings, before moving into specific subsections. Similarly, I would recommend first focusing on the main study aims (i.e. exploring the impact of stress, psycap), rather than the control variables. In my view, the subheadings make the discussion harder to follow, as there is currently overlap between the sections.
Response : Thank you for the comments. First, we would like to thank you for noticing the control variables in our study. By adding control variables in models, it can increase the explanary power of predicting variables (PsyCap) to dependent variables (athletic burnout). Besides, we could also detect whether those control variables caused athletic burnout. Hence we could draw some practical suggestions to athletes and coaches. In our study, the results showed one control variable- training days per week-had sinificant positive effects on athletic burnout. We found it important to make suggestions to coaches, since to our knowledge (as Physical Education professors), most athletes have 4-6 days’ regualr training schedule per week. We would like to suggest coaches to decrease athletes’s weekly training days.
2nd review: I appreciate that the inclusion of control variables is relevant, however, the response provided here does not address the concerns outlined in my comment. As outlined previously, I feel the discussion would benefit from providing a broader initial discussion of the key findings before moving into specific subsections, and that the focus should be on the main hypotheses before focusing on control variables.
1st Review. There is currently no discussion of the limitations of the work. It is important for the authors to acknowledge relevant limitations, and outline recommendations for future work.
The manuscript would also benefit from a final concluding paragraph, as it currently ends quite abruptly.
Response: Thank you for the comments. We revised the conclusion section and added 5.3 Implications for future reserch.Thank you.
2nd review: These additions address some of the concerns raised and improve the piece. However, there is still no discussion of the limitations of the work.
L320-325: The argument here is again lacking relevant references, and also seems to go beyond the scope of the current study. Specifically, considerations around, training schedules, injuries and overtraining do not necessarily relate to the idea of stress or contest pressure. Instead, it may be useful to consider to explore potentially other relevant factors, such as coach-created climate, and interventions focused on stress-reduction, such as mindfulness-based approaches.
Response: Thank you for the suggestions. We made suggestions accordingly
2nd review: It is difficult to identify where these changes have been made as there is limited detail provided here.
Additional Comments:
Abstract, L26: The impact of days of training per week on burnout is described as negative, but the beta value included is positive.
Introduction, L65: A reference should be included to support this statement.
L80: A reference is needed to support the claim about the popularity of baseball in Taiwan.
L111 – L116 Hypothesis should be worded in the future tense
L210 – L213 A reference should be included to support this statement
L230 – L231: The hypothesis states that life stress will have a negative impact on burnout, which is contrast to the last line from the preceding paragraph. Based on existing research, it would be likely thar stress would positively predict burnout.
L237 – L241. There is repetition with the informaiton reported in-text and in Table 1. This should be avoided where possible.
L259 – 260: It would be useful to provide some more detail on how questions were rephrased, and some additional rationale for why this was necessary
Section 2.2: It would be useful to provide information on the reliability of measures used in similar populations to support their use in this study.
L355 – 357: The claim that most students want to play professionally should be supported by a relevant reference.
L365 – L376: These claims should be supported by relevant references
L370 – 375: It would be useful to draw on existing research/intervention studies exploring how to improve PsyCap. Currently, the suggestions made are beyond the scope of this study and are not supported by references to other work.
L378– L382: The claims made across this section should be supported by relevant references
L399 – L402: The suggestions made around quality time away from sport are beyond the scope of this study and are not supported by references to other work.
L419 – 422: These claims should be supported by relevant references
L427 – L431: These claims should be supported by relevant references
L470 – L476: These claims should be supported by relevant references
L478 – L479: The language used here (“so coaches, come on, face the reality, the training is just too many”) is journalistic style, and would benefit from a more academic-writing style.

Typographic and grammatical errors are evident throughout, and the manuscript would be strengthened by efforts to address these.
Author Response
Reviewer 1
I would like to thank the authors for considering and addressing some of the comments and concerns raised in response to the previous submission. Where changes have been made, the work has been improved. However, it was notable that a substantial number of comments were not addressed in-text, and in some cases no response or rationale was provided in the accompanying document. Specifically, it is my opinion that the following comments were not addressed, either at all or sufficiently well;
Point 1:
1st Review, Abstract: The opening line in the abstract suggests that training demands will be explored as a predictor of stress and burnout, and this is not the case. I would recommend focusing on introducing the concept of stress more explicitly in terms of its relationship to burnout, as you do for PsyCap in the subsequent sentence. Response 1: Thank you for the comments. We revised the aims of the study as the following: “the purpose of this study was to explore the influences of university baseball athletes’ psychological capital on their life stress and burnout and provide practical suggestions for athletes and coaches to reduce their life stress and burnout.” To make our study goal clearer. Please refer to lines 16- 18.Thank you very much.
2nd review: The concern raised in the original comment has not been addressed, as the opening line (L14 – L15) has not been amended as suggested.
Point 2
1st Review, Introduction: It would be useful to expand on the description burnout provided to include a definition of each of the three burnout dimensions. In addition, as two different definitions of burnout are provided (i.e. Eades’ and Radeke’s definitions), it should be made clear here which definition will be employed moving forward in the study. To this end, I suggest that Raedeke’s definition is most appropriate, as it is the most commonly employed definition in the athlete burnout literature (e.g. Gustafsson et al., 2017)
Response 3: Thank you for the comments. The study mainly focused on athletic burnout and we just focused on the definition of athletic burnout.
2nd review: The concern raised in the original comment has not been sufficiently addressed - Two different definitions of burnout are introduced and the authors need to be clear on which definition they have employed in the study.
Response: We adapt Raedeke’s definition. Please refer to Ln #40-42. Thank you.
Point 3
1st Review: The introduction to the concept of stress seems to move back and forth between mental health and stress. In my opinion, this paragraph would benefit from a more focused discussion of stress in the context of sport, and its relationship to burnout and psychological capital specifically. Currently, much of this information comes in sections 1.2 and 1.4, but I feel that integrating it into the main body of the introduction would contribute to a more logical flow in this section. As it stands, concepts are being introduced with minimal context, which is confusing for the reader.
2nd review: This point has not been addressed in-text and the author has not provided any direct response.
Response: Thank you for comment. To avoid confusing, we deleted the paragraphs relating to mental health and revised this section and the following sections. The main focus of these sections are to relate 1.life stress with athlete burnout; 2. PsyCap with life stress and athlete burnout.
Point 4
1st Review: Currently, the main body of the introduction also provides limited insight into our understanding of the relationship between PsyCap and burnout, and I think the following recommendations may help to address this;
1. A definition of the core components of psychological capital would be a useful addition to the introduction. Currently, self-efficacy, hope, optimism and resilience are mentioned but not
defined in this section, with the first definition of self-efficacy is provided in the integrations section. This information would be much more useful earlier in the manuscript.
2nd review: This point has not been addressed in-text and the author has not provided any direct response.
Response: Thank you for the comment. We added definitions for PsyCap dimensions. Please refer to Ln #69-72.
Point 5
L320-325: The argument here is again lacking relevant references, and also seems to go beyond the scope of the current study. Specifically, considerations around, training schedules, injuries and overtraining do not necessarily relate to the idea of stress or contest pressure. Instead, it may be useful to consider to explore potentially other relevant factors, such as coach-created climate, and interventions focused on stress-reduction, such as mindfulness-based approaches.
Response: Thank you for the suggestions. We made suggestions accordingly
2nd review: It is difficult to identify where these changes have been made as there is limited detail provided here.
Response: Thank you for the comment. In section 4.2 we discussed the relationships of Sports PsyCap , life stress and athletic burnout. Ln#340-360 did address some intervention courses to increase athlete PsyCap.
Point 6
Abstract, L26: The impact of days of training per week on burnout is described as negative, but the beta value included is positive.
Response: Thank you. We added “-“ accordingly.
Point 7
Introduction, L65: A reference should be included to support this statement.
L80: A reference is needed to support the claim about the popularity of baseball in Taiwan.
Hsieh, S.-Y. The birth of Kokukyū precedent: History of Taiwan Ball in the Nikji period, 2012, National Museum of Taiwan History, Tainan, Taiwan.
Point 8
L111 – L116 Hypothesis should be worded in the future tense
Response: Thank you for suggestions. We revised them accordingly.
Point 9
L230 – L231: The hypothesis states that life stress will have a negative impact on burnout, which is contrast to the last line from the preceding paragraph. Based on existing research, it would be likely that stress would positively predict burnout.
Response: Thank you for the comment. Please refer to Ln#187-190.
Point 10
Section 2.2: It would be useful to provide information on the reliability of measures used in similar populations to support their use in this study.
Response: Thank you for the comment. The reliability of the measurements were tested using PLS-SEM. Please to section 3.1 (Ln#235-251).
Point 11
L355 – 357: The claim that most students want to play professionally should be supported by a relevant reference.
Response: Thank you for the comment. We added a reference to support this claim. Please refer to Ln 304.
Point 12
L370 – 375: It would be useful to draw on existing research/intervention studies exploring how to improve PsyCap. Currently, the suggestions made are beyond the scope of this study and are not supported by references to other work.
Response: Thank you for the comment. We added a reference.
Stajkovic , A.D., Luthans, F. Social cognitive theory and self-efficacy: Going beyond traditional motivational and behavioral approaches, Organizational Dynamics,1998, 26(4), 62-74,doi:10.1016/S0090-2616(98)90006-7
Reviewer 2 Report
The authors use the concept of psychological capital throughout the text but work with the dimensions of positive psychological capital. The authors could and should revise the work replacing, from the title, the term used by positive psychological capital in order to better elucidate the readers.
It is fantastic that the study includes sociodemographic variables as predictors. However, the hypotheses raised need more foundation. In addition, the hypotheses do not present the expected outcome. For example, in the hypothesis: "College baseball athletes’ grade had influences on athletic 111 burnout", but how? Low college baseball athletes’ grades increase burnout, or high college baseball athletes’ grades increase burnout due to the effort used in two activities. All hypotheses are interests and pertinent but need better argumentation.
Also, the discussion could be better, particularly in point 4.1. We do not find a profound reflection on results, especially when the results were not expected and predicted by the previous literature. However, these results are the most interesting in the study. The authors should improve this section.
The conclusion of the work is unclear and little congruent with the discussion. Should be totally reviewed.
Author Response
Point1: The authors use the concept of psychological capital throughout the text but work with the dimensions of positive psychological capital. The authors could and should revise the work replacing, from the title, the term used by positive psychological capital in order to better elucidate the readers.
Response: Thank you for the comment. We had revised the sections related to psychological capital as (please refer to line 67) “The concept of Psychological capital (PsyCap) was first developed in the 2000s within the positive psychology movement by Luthans, Youssef & Avolio [1]. They de-fined PsyCap as "an individual's positive psychological state of development". Namely, PsyCap refers to our mental resources and their ability to help us get through tough situations. Well-developed PsyCap is reported to have benefits for both individuals and organizations. According to Luthans and Larson [2], people with greater PsyCap typically enjoy higher levels of job satisfaction, commitment, and overall wellbeing. In a study of PysCap’s influences on job performances and stress, Abbas and Raja found that people with higher levels of PsyCap can be linked to lower stress and better performances [3]. “
References
- Luthans, F.; Youssef, C.M.; Avolio, B.J. Psychological capital. Oxford, UK: Oxford University Press, 2007.
- Luthans, F.; Larson, M. Potential Added Value of Psychological Capital in Predicting Work Attitudes, Journal of Leadership & Organizational Studies, 2006, 13(1), 1-27.
- Abbas, M. ; Raja, U. Impact of Psychological Capital on Innovative Performance and Job Stress, Canadian Journal of Administrative Sciences, 2015, 32(2), 128-138.
In this way, we believe readers can have a better idea of what psychological capital is and we stick to psychological capital in our title. Thank you.
Point 2: It is fantastic that the study includes sociodemographic variables as predictors. However, the hypotheses raised need more foundation. In addition, the hypotheses do not present the expected outcome. For example, in the hypothesis: "College baseball athletes’ grade had influences on athletic burnout", but how? Low college baseball athletes’ grades increase burnout, or high college baseball athletes’ grades increase burnout due to the effort used in two activities. All hypotheses are interests and pertinent but need better argumentation.
Response:
Thank you very much for the suggestions. First of all, when we investigate the causal effect of a treatment/action/intervention with non-experimental data, we need to “control for” confounding variables. Controlling for a variable means estimating the difference in average outcome between a treatment group and a control group within a specific category/value of the controlled variable.
Initially, we wanted to control the athletes’ background variables so we could better estimate the influences of PsyCap on life stress and athletic burnout. Later on, we read some article relating to athletes’ background information (grade, years of training experience and training frequency) and found interesting results. We therefore added these influences in our hypotheses. However, we didn’t mentioned the referred articles in our study. We apology for this. So thank you for the reminder. This time, we added those related articles and revised our hypotheses in the following (please refer to line99-108).
Some studies had used demographical variables as control variables to examine the influences of independent variables to predicted variables [17, 18]. Reasons for that were to control the variables so to better estimate the independent variables. Some studies also indicated that several control variables are related to athletic burnout. For example, different grades’ athletes exhibited different level of burnout [19, 20]. Speciically, Yang et al. [20] reported that in junior high school, lower grade’s athletes had higher athletic burnout than higher grade’s athletes. As for training experience, studies suggested that athletes with different training experience exhibit different level of burnout [21, 22, 23, 24]. For example, Yang et al. [22] pointed out that female college baseball players with greater training experience had higher burnout. As for training frequency, athletes with different training frequency exhibit different level of burnout [23, 24, 25]. For example, Cheng and Yang [23] pointed out that Judo players with greater training days per week had higher burnout. Based on above-mentioned studies, the following hypotheses are proposed:
Hypothesis 1-1 (H1-1): College baseball athletes’ grade have negative influences on athletic burnout.
Hypothesis 1-2 (H1-2): College baseball athletes’ years of training experience have positive influences on athletic burnout.
Hypothesis 1-3 (H1-3): College baseball athletes’ training days per week have negative influences on athletic burnout.
Point3: The discussion could be better, particularly in point 4.1. We do not find a profound reflection on results, especially when the results were not expected and predicted by the previous literature. However, these results are the most interesting in the study. The authors should improve this section.
Response: Thank you for the comment. We added supplementary descriptions in this section. Please refer to the following:
As for demographic variables, our results showed that grade and years of training experience had no influences on athletic burnout which are not consistent with previous studies [20, 22]. As for grade, Yang et al. pointed out junior high school student athletes’ grade was relating to burnout negatively [20]. As for junior high school athletes, their physical fitness and motor skills vary with grade. Junior high school students undergo physiological growth. As they get older, their physical status grow to mature status. Besides, they get more improvements in motor skills from training. Therefore, younger athletes might have more athletic burnout than older athletes. On the other hand, college athletes’ physical fitness and motor skills are pretty much the same and stable. Therefore, grade may not be a control variable for athletic burnout. As for years of training experience, the study found college baseball players’ training experience had not influences on athletic burnout which is different from Yang et al. [22] in which the study subject is female players which this study focuses on male players. Therefore, it can be concluded that male college baseball players’ years of training experience has no influences on athletic burnout. As for training days per week, our study found the training days per week had negative influences on athletic burnout. This is consistent with some previous studies [23, 48]. According to descriptive statistics, 43.5% of athletes had 5 or 6 days training per week. Since baseball is a very competitive sport item in Taiwan, most baseball student athletes desire to join professional baseball teams[]. Hence training is intensive. However, as McGuff and Little [48] pointed out that the best training approach is “high intensity and low frequency”, in that way, athletes get to enough rest for recovery. Five or six-day’s training per week might cause athletes’ physical and mental burden hence induce stress and fatigue [49]. Therefore, the better approach is to adopt structured training plans, which comprise of high intensity, low frequency training. In that way, athletes are allowed to have ade-quate resting and recovery periods and athletic burnout can be reduced.
Point 5:The conclusion of the work is unclear and little congruent with the discussion. Should be totally reviewed.
Response: Thank you for the suggestion. We revised the section as the following:
The study explored the influences of college baseball players’ PsyCap (self-efficacy, hope, optimism and resilience) on life stress and athletic burnout using athletes’ grade, years of training experience and training days per week as control variables. The study found self efficacy, hope and optimism had negative influences on life stress. Hope had negative influences on athletic burnout. Also, life stress had positive influence on athletic burnout. As for demographical variables, the study found athletes’ training days per week had negative influences on athletic burnout.
Reviewer 3 Report
I appreciate the invitation to review this article. The research is interesting as it addresses a topic of importance for athletes and the circumstances surrounding them. Here are my suggestions to the authors with the aim of improving the article:
- I believe it is necessary to clearly state the research objective. While the hypotheses are present, it is important to explicitly specify the objective of the conducted research. I suggest including it at the end of the introduction.
- In section 2.3 "Data Analysis," I recommend that the authors provide more details, including the specific statistics used and the significance levels of the p-values.
- It would be beneficial for the authors to provide a more in-depth justification for the use of the structural equation modeling (SEM) framework. For example, they could explain why no relationships were introduced between "Self efficacy," "hope," and "optimism." In summary, the analysis employs a confirmatory factor analysis model utilizing structural equations.
- In the proposed model (Fig. 1), it is advisable to include a global model fit statistic, such as RMSEA.
- Additionally, I would like to highlight the sample size in the study. Despite its heterogeneity, I consider it sufficient to support the conclusions.
Author Response
I appreciate the invitation to review this article. The research is interesting as it addresses a topic of importance for athletes and the circumstances surrounding them. Here are my suggestions to the authors with the aim of improving the article:
Point 1: I believe it is necessary to clearly state the research objective. While the hypotheses are present, it is important to explicitly specify the objective of the conducted research. I suggest including it at the end of the introduction.
Response: Thank you for the comment. The research objective was described in lines 91-92 as the following:
Therefore, the aims of this study were to use sports PsyCap as a predictor to examine its influences on college baseball athletes’ life stress and burnout.
Point 2: In section 2.3 "Data Analysis," I recommend that the authors provide more details, including the specific statistics used and the significance levels of the p-values.
Response: thank you for the comment. We had explained why PLS was used for hypothesis testing as the following:
The descriptive analyses were performed using SPSS for windows and the tests of hypotheses were performed using partial least square (PLS) structural equation modeling by using WarpPLS 8.0 developed by Kock [46]. According to Chin [48], PLS can be used for exploratory and confirmatory analyses. PLS benefits from: (1) being distributionfree, (2) requiring only a small sample size, (3) having the ability to process multiple dependent and independent variables simultaneously,(4) handling collinearity, and (5) processing both formative or reflective indicators. In the study hypotheses include multiple independent and dependent variables, therefore, PLS employed for hypothesis testing. A significant level of a = 0.05 is used in the study.
Point 4: It would be beneficial for the authors to provide a more in-depth justification for the use of the structural equation modeling (SEM) framework. For example, they could explain why no relationships were introduced between "Self efficacy," "hope," and "optimism." In summary, the analysis employs a confirmatory factor analysis model utilizing structural equations.
Response: Unlike traditional regression analysis, SEM can estimate all path coefficients in model and are more accurate than regression analysis. In addition, SEM can handle measurement errors. However, the tool can only provide the test results but cannot explain why the relationships are significant or not. We do discuss results in the discussion section. In this section, we only described results.
Point 5:In the proposed model (Fig. 1), it is advisable to include a global model fit statistic, such as RMSEA.Additionally, I would like to highlight the sample size in the study. Despite its heterogeneity, I consider it sufficient to support the conclusions.
Response: Thank you for the comments. We used partial least squares structural equation model for data analysis. The analysis is very different from traditional covariances-based SEM. Please refer to the following for detail: Hair, J. F., Hult, G. T. M., Ringle, C. M., & Sarstedt, M. (2022). A Primer on Partial Least Squares Structural Equation Modeling (PLS-SEM), 3rd ed. Thousand Oaks, CA: Sage. The outputs from PLS-SEM do not provide AGI, NFI or RMSEA. Besides, PLS-SEM analysis can treat data regardless of distribution and it can deal with small sample size. As for sample size (n=428), it is revealed in the “2.1 participant” section (Ln # 99). Thank you.
Round 2
Reviewer 1 Report
The authors should be commended on the improvements made to the work since the original submission, but some issues remain unaddressed or, in my opinion, have not been addressed sufficiently. Particular effort should be taken to ensure broad, unsupported claims are avoided, and recommendations do not go beyond the scope of the work.
Abstract:
The below issue raised in the both the first and second rounds of review has still yet to be addressed;
1st Review, Abstract: The opening line in the abstract suggests that training demands will be explored as a predictor of stress and burnout, and this does not appear to be the case, as training load is included as a predictor of burnout only. I would recommend focusing on introducing the concept of stress more explicitly in terms of its relationship to burnout, as you do for PsyCap in the subsequent sentence.
2nd review: The concern raised in the original comment has not been addressed, as the opening line (L14 – L15) has not been amended as suggested
L25: The previous review highlighted that the impact of days of training per week on burnout is described as negative, but the beta value included is positive. The authors have no included a negative beta value, but this is at odds with the positive beta value included in figure 1. It is my understanding that training days per week was positively associated with burnout, and this result should be reported clearly in the abstract.
L44: While stress has been identified as a key predictor of burnout, research has also pointed to a number of other important factors, and thus I do not believe this statement is accurate. Furthermore, the supporting reference provided (i.e. Gustafsson et al., 2017) also highlights additional factors beyond stress. I recommended rephrasing this sentence to reflect this.
L105: Hypothesis 1-3 states that training days will have a negative influence on athletic burnout. This is in contrast to the research referenced in the preceding, which suggests that greater training frequency is associated with higher burnout (i.e. positive relationship)
L311 and L323: Both the “Sports PsyCap and life stress” and the “Sports PsyCap, life stress and athletic burnout” subsections are labelled 4.2
L318 – L322: It would be useful to provide references for similar intervention approaches, as the current recommendations are somewhat beyond the scope of this work.
L326 – L329: these statements are lacking relevant references. The statement “That hope keeps them going” is also quite sweeping, and doesn’t account for the myriad of reasons why athletes participate in sport
L335 – L339: The focus on social isolation appears to be a jump from discussion of the current study’s focus on hope. As such, recommendations to spend time with others is beyond the scope of the current study and relevant supporting references should be provided.
L384 – L392 Please include relevant references to support statements made across these sentences.
Sentence syntax and grammar could be improved in some places.
Author Response
The authors should be commended on the improvements made to the work since the original submission, but some issues remain unaddressed or, in my opinion, have not been addressed sufficiently. Particular effort should be taken to ensure broad, unsupported claims are avoided, and recommendations do not go beyond the scope of the work.
Response: Thank you so very much for carefully reviewing our article. Your efforts are highly appreciated. Thank you very much.
Abstract:
The below issue raised in the both the first and second rounds of review has still yet to be addressed;
1st Review, Abstract: The opening line in the abstract suggests that training demands will be explored as a predictor of stress and burnout, and this does not appear to be the case, as training load is included as a predictor of burnout only. I would recommend focusing on introducing the concept of stress more explicitly in terms of its relationship to burnout, as you do for PsyCap in the subsequent sentence.
2nd review: The concern raised in the original comment has not been addressed, as the opening line (L14 – L15) has not been amended as suggested
Response: Thank you for the comment. We deleted the opening statement.
L25: The previous review highlighted that the impact of days of training per week on burnout is described as negative, but the beta value included is positive. The authors have no included a negative beta value, but this is at odds with the positive beta value included in figure 1. It is my understanding that training days per week was positively associated with burnout, and this result should be reported clearly in the abstract.
Response: Thank you for the comment. Yes, you are right. Training days per week have positive influences on athlete burnout. We had corrected this mistake in abstract. Thank you so very much.
L44: While stress has been identified as a key predictor of burnout, research has also pointed to a number of other important factors, and thus I do not believe this statement is accurate. Furthermore, the supporting reference provided (i.e. Gustafsson et al., 2017) also highlights additional factors beyond stress. I recommended rephrasing this sentence to reflect this.
Response: Thank you for the comment. This time we use “Smith RE: Toward a cognitive–affective model of athletic burnout. J. Sport Psychol. 1986, 8:36-50.”
(The most cited paper on athlete burnout and the most influential burnout model.) as a reference to support our statement. Gustafsson et al. (2017) pointed out according to self-determination theory (STD), burnout is associated with motivation. As Smith’s study pointed out that athlete burnout developed through a stress-based process influenced by personal and motivational factors. Therefore, the study used stress as a cause for athlete burnout. Please refer to Ln 42-43. Thank you.
L105: Hypothesis 1-3 states that training days will have a negative influence on athletic burnout. This is in contrast to the research referenced in the preceding, which suggests that greater training frequency is associated with higher burnout (i.e. positive relationship)
Response: Thank you for the comment. We revised it.
L311 and L323: Both the “Sports PsyCap and life stress” and the “Sports PsyCap, life stress and athletic burnout” subsections are labelled 4.2
Response: Thank you for the note. We corrected it.
L318 – L322: It would be useful to provide references for similar intervention approaches, as the current recommendations are somewhat beyond the scope of this work.
Response: Thank you for the comment. We added two references into this section. Please refer to Ln 318-322.
L326 – L329: these statements are lacking relevant references. The statement “That hope keeps them going” is also quite sweeping, and doesn’t account for the myriad of reasons why athletes participate in sport.
Response: Thank you for the comment. The sentence was deleted.
L335 – L339: The focus on social isolation appears to be a jump from discussion of the current study’s focus on hope. As such, recommendations to spend time with others is beyond the scope of the current study and relevant supporting references should be provided.
Response: Thank you for the comment. We deleted this part and revised this section. Please refer to Ln 323-332. Thank you.
L384 – L392 Please include relevant references to support statements made across these sentences.
Response: Thank you for the comment. We added the following articles as references.
- Wang, J.-X. Analysis of psychological capital value of excellent athletes and intervention strategies, Journal of Sports Adult Education, 2010, 26(2), 56-59.
- Wang, T.W.; Lee, W.P.; Lu, J.H. How coping resource reduces athletes’ burnout in sport settings: The mediating role of life stress. Physical Education Journal, 2015, 48(3), 251-264. doi:10.3966/102472972015094803003